# Postpartum Tobacco Use and Perceived Stress among Alaska Native Women: MAW Phase 4 Study

**DOI:** 10.3390/ijerph16173024

**Published:** 2019-08-21

**Authors:** Christi A. Patten, Kathryn R. Koller, Christie A. Flanagan, Vanessa Hiratsuka, Zoe T. Merritt, Flora Sapp, Crystal D. Meade, Christine A. Hughes, Paul A. Decker, Neil Murphy, Timothy K. Thomas

**Affiliations:** 1Department of Psychiatry and Psychology, Mayo Clinic, 200 First St. SW, Rochester, MN 55905, USA; 2Clinical and Research Services, Division of Community Health Services, Alaska Native Tribal Health Consortium, 4000 Ambassador Dr., Anchorage, AK 99508, USA; 3Southcentral Foundation, Department of Obstetrics and Gynecology, Alaska Native Medical Center, 4320 Diplomacy Dr., Ste. 1800, Anchorage, AK 99508, USA; 4Department of Health Sciences Research, Mayo Clinic, 200 First St. SW, Rochester, MN 55905, USA

**Keywords:** postpartum, smoking, tobacco, stress, Alaska Native, women

## Abstract

Prior research explored reasons for tobacco use among pregnant Alaska Native (AN) women but did not address the postpartum period. This study followed up with AN women one to three years postpartum who had participated in a prenatal smoking cessation intervention study (Motivate Alaska Women (MAW) Phase 3) and had consented to be re-contacted for future studies. Of 47 eligible women, 32 (68%) participated. A semi-structured phone interview was conducted a mean of 2.0 years after delivery (range 1.6–2.8). Measures assessed self-reported tobacco use status in the 12 months after delivery, at 12 months postpartum, and at the time of the interview; reasons for maintaining abstinence, continued use, or relapse; and included the Perceived Stress Scale (PSS) and Negative Affect (NA) scale. Content analysis was used to generate themes from open-ended response items. Tobacco use was reported by 23 women (72%) at delivery, 30 (94%) within the 12 months after delivery, 27 (84%) at 12 months postpartum, and 29 (91%) at the time of the interview. Among nine women not using tobacco at delivery, seven (78%) relapsed during the 12 months after delivery. Of the 29 current tobacco users, 28 (97%) smoked cigarettes. Twenty-seven participants (84%) reported stress and 15 (52%) indicated addiction as reasons for continuing, starting, or resuming tobacco use. Types of stressors were related to parenting and traumatic experiences. Among current tobacco users, mean NA score (18.7) was significantly higher (*p* = 0.01) than the normative mean (14.8), but no differences were detected for PSS score. In this sample of AN women, postpartum tobacco use was highly prevalent, and stress was a primary reason that women endorsed for using tobacco. These preliminary results have several practice and research implications for exploring ways to support non-tobacco use among postpartum AN women.

## 1. Introduction

Among Alaskans, the prevalence of any tobacco use before pregnancy was two-fold higher among Alaska Native (AN) women than among white women (60% vs. 28%), and the prevalence of any tobacco use during pregnancy (43% vs. 14%) and at two to nine months postpartum (53% vs. 19%) were each nearly three-fold higher [1]. In addition, postpartum relapse to cigarette smoking among women who had quit during pregnancy was significantly greater among AN women compared to white women (57% vs. 41%) [1]. Prior studies documented that tobacco is not used for traditional (ceremonial or religious purposes) among AN people [2]. Postpartum smoking is associated not only with increased morbidity and mortality for the mother, but exposure to second-hand smoke has adverse health effects on lung growth and development among infants and young children [3]. Few effective interventions address postpartum smoking relapse generally [4], and none have been evaluated among American Indian or Alaska Native (AI/AN) women. Moreover, no previous work assessed reasons for tobacco use among postpartum AI/AN women.

With few exceptions [5], higher levels of perceived stress, stressful life events, and depression are associated with smoking during pregnancy [6,7,8,9], as well as postpartum smoking relapse [10,11] in the U.S. and other countries. In a qualitative study conducted in the rural Yukon–Kuskokwim Delta region with 60 pregnant AN women, family members, and Elders, participants reported stress was the major reason why women use tobacco during pregnancy and that programs to reduce tobacco use should address stress [12]. During our research team’s previous biomarker study, we enrolled 118 AN pregnant women from the Anchorage area (54 smokers, 64 non-smokers) and assessed reasons for smoking during pregnancy [13]. The most common reasons reported were addiction (74% overall; 63% of smokers, 53% of nonsmokers) and stress/affect management (73% overall; 69% of smokers, 56% of nonsmokers). However, prior studies did not address reasons for quitting, continued tobacco use, or relapse during the postpartum period among AN women. A systematic review of smoking in pregnancy among Indigenous women in four countries including Australia, New Zealand, Canada, and the U.S. likewise highlighted gaps in the knowledge on barriers, motivators, and cultural strengths supporting quitting tobacco to develop more effective approaches [14].

The Biomarker Feedback to Motivate Tobacco Cessation in Pregnant Alaska Native Women (MAW) study was a three-phase study conducted in Anchorage, Alaska, with the overall goal to develop and pilot test an intervention using biomarkers of nicotine and tobacco-specific carcinogens to increase motivation of pregnant AN women to quit smoking. Phase 1 established a moderate to strong correlation between maternal urine cotinine (metabolite of nicotine) concentrations and infant urine tobacco-specific N-nitrosamine levels [15]. In Phase 2, qualitative feedback was obtained in a different sample on acceptability of the information, and how best to present it in a tobacco cessation intervention tailored for pregnant AN women [16]. Using results from Phase 2 as a guide, we developed a culturally-tailored biomarker feedback intervention, which was then piloted in Phase 3 in a new sample of pregnant AN women using a randomized, two-group design [17]. Sixty pregnant AN women were enrolled and randomly assigned to receive a five-week intervention consisting of the Southcentral Foundation (SCF) Quit Tobacco Program standard of care for tobacco cessation (*n* = 30) or the same standard of care plus personalized biomarker feedback information to inform about fetal exposure to tobacco-specific carcinogens (*n* = 30). The biochemically verified smoking abstinence rates at delivery were identical for both study conditions (20% for both groups) [17].

For the current study, we added a fourth phase (MAW Phase 4) to follow-up with women one to three years postpartum who participated in Phase 3 and had consented to be re-contacted for future studies. Our aims were to: (1) assess tobacco use status during the first 12 months after delivery, at one year postpartum, and current tobacco use at the time of the follow-up interview, with quit status previously obtained at delivery; (2) explore risk factors (e.g., stress) for continued use and relapse, and resilience factors (e.g., family support) factors for abstinence; and (3) compare levels of perceived stress and negative affect among current tobacco users versus non-users.

## 2. Methods

The fourth phase of the MAW study was reviewed and approved separately from the first three phases by the Mayo Clinic and Alaska Area Institutional Review Boards. The Alaska Native Tribal Health Consortium and Southcentral Foundation provided Tribal approval. A study-specific community advisory board was formed for the MAW study. The MAW Phase 3 trial was registered with clinicaltrials.gov (NCT02431611). Participants in the MAW Phase 3 study were enrolled between March 2015 and July 2016. Data collection for the MAW Phase 4 postpartum follow-up study occurred between March 2018 and November 2018.

### 2.1. Participants

To be eligible for Phase 4, the woman must: (1) have participated in Phase 3 of the MAW study [17] and (2) previously consented to being contacted in the future about participation in other studies related to tobacco use and health. Of the 60 women who enrolled in Phase 3 of the MAW pilot study, 47 (78%) had consented to be re-contacted and were thus eligible for this follow-up study.

Study staff contacted women who were eligible to participate by mail and/or phone using the contact information previously collected in MAW Phase 3. These women were asked if they would like to participate in a study to assess smoking status beyond delivery. If they were interested in participating, verbal consent was conducted over the phone or in person prior to administering the assessment. For participants whose contact information changed since their participation in the MAW Phase 3, study staff updated contact information obtained from the health record. Study staff attempted to contact women within one to two years after delivery. However, after several attempts, study staff was unable to contact several women, and made multiple attempts to reach women who eventually participated. This resulted in a larger time window (1–3 years postpartum) than planned for assessing current tobacco use status.

### 2.2. Measures

#### 2.2.1. Socio-Demographic and Tobacco Use Characteristics

Baseline participant demographic and tobacco use characteristics were obtained from the MAW Phase 3 study [17]. Socio-demographic information included: age, married status, education, number of weeks at gestation, number of biological children, spouse/partner smoking status, presence of home smoking ban (including arctic entry), hours exposed to cigarette smoke each day, and living with other smokers. MAW Phase 3 baseline tobacco use characteristics obtained were Contemplation Ladder score (readiness to quit) [18], cigarettes smoked per day, Fagerström Test for Cigarette Dependence (FTCD) score [19], and urinary cotinine concentration.

#### 2.2.2. Phase 4 Interview

Study staff trained in both qualitative and quantitative data collection administered a structured phone-based interview.
Social-Environmental Characteristics. Women were asked if their spouse/partner used tobacco with response options yes or no. Participants were asked to report how many hours per day they see or smell cigarette smoke. In addition, participants were asked “what are the rules about smoking inside your home?” with response options being: no one is allowed to smoke anywhere inside the home, smoking is allowed in some rooms or at some times, and smoking is permitted anywhere.Tobacco Use. The interviewer first noted the participant’s self-reported tobacco use status at delivery that was obtained from the MAW Phase 3 study [17]: “according to the information we collected, at the time of delivery you were using tobacco/not using tobacco.” Participants were then asked about their tobacco use status at three time points: (1) during the first 12 months postpartum: “between the time you delivered and 12 months after (child’s first birthday), have you used tobacco?” and “did you try to quit during the 12 months after your delivery?”; (2) at 12 months postpartum: “at one year after your delivery, were you using tobacco?” and (3) at the time of the interview (current tobacco use status): “have you used tobacco in the last 7 days?” [20]. The types of tobacco/nicotine products assessed were: cigarettes, electronic cigarettes, Iqmik/buluq/peluq/blackbull (a homemade form of smokeless tobacco [21]); Copenhagen or other chewing tobacco, and other (open-ended response). Tobacco/nicotine use status was not biochemically verified. To assess cigarettes smoked per day (CPD), current cigarette smokers were asked “in the past week, how many cigarettes, on average, did you smoke per day?”Reasons for Continuing, Resuming, or Quitting Tobacco Use. Participants who currently used tobacco were asked “what are the reasons you continue to use tobacco?” Those who had quit during the 12 months after delivery but then resumed smoking were asked “what are the reasons you started using tobacco again?” Participants who had quit tobacco were asked “what are the reasons you quit using tobacco?” The response format for these questions was open ended. All participants were asked “what do you think are reasons a woman would start using tobacco again after delivering her baby?” [18]. The interviewer read this question, waited for and recorded any response(s) using a checklist, and then prompted for any other reasons [13]. Response options on the checklist were: because it’s safe to use, other women I know use it, control stress, manage depression, boredom, feeling tired (fatigue), to feel better, to avoid or limit alcohol use, suppress hunger, and other. The interviewer wrote in any other additional responses.Perceived Stress. All participants completed the validated four-item Perceived Stress Scale (PSS) [22]. Items are rated in reference to the past month using a five-point scale ranging from 0 = never, 2 = sometimes, to 4 = fairly often. Examples are: “how often have you felt that you were not able to control the important things in your life?” and “how often have you felt problems were piling up so high that you could not handle them?” Two of the items are reverse coded and the total score is calculated by summing across all four items. Possible scores can range from 0–16 with higher scores indicating greater levels of perceived stress. The normative mean score was 4.49 in a U.S. probability sample [23].Negative Affect. Negative affect (NA) was assessed using the validated Positive and Negative Affect Schedule (PANAS) [24,25]. Ten items (e.g., irritable, distressed) are rated on a five-point scale of 1 (very slightly or not at all) to 5 (extremely). Possible scores range from 10–50 with higher scores indicating greater negative affect. The mean score in a representative sample was 14.8 [25].

### 2.3. Statistical Methods

Participant demographic characteristics obtained from the MAW Phase 3 study were summarized using descriptive statistics and compared between Phase 3 study groups (intervention and control group) using the chi-square test for categorical variables and the two-sample rank sum test for continuous variables. Social environmental characteristics, tobacco use status, PSS scores, NA scores, and reasons for tobacco use in pregnancy from the checklist obtained at the time of the interview were summarized using descriptive statistics. A one-sample *t*-test was used to compare the mean for this sample on PSS and NA scores compared to the means observed among representative U.S. samples. There were not sufficient numbers of non-users to compare these scores with current tobacco users. *P* values of < 0.05 were used to denote statistical significance. Participant responses on open-ended questions assessing reasons for continuing or relapse to tobacco use were recorded by study staff on the interview form as direct quotes or paraphrases and summarized using content analysis [26]. Two independent raters coded the data with inter-rater agreement of 97.6%. Discrepancies were discussed until consensus was reached. For all analyses, we suppressed reporting numbers for cell sizes <5 to protect participant confidentiality and minimize risk of exposing identities in this relatively small population.

## 3. Results

### 3.1. Participants

Figure 1 shows the participant flow. Of the 47 eligible women, 32 (68%) participated. The remaining women were lost to follow-up due to inability to contact, incarceration, or were deceased. Participants were interviewed at a mean of 2.0 years after delivery (standard deviation (SD) = 0.34, range 1.6–2.8), with interviews ranging from 20–40 min in duration. Fourteen respondents had been randomly assigned to the biomarker feedback condition and 18 to the control condition in MAW Phase 3.

Table 1 presents the participant baseline characteristics assessed at time of enrollment to the MAW Phase 3 study. No statistically significant differences were detected between study groups (data not shown due to many cell sizes <5). The baseline characteristics were similar to the overall sample of 60 women who enrolled in the MAW Phase 3 study [16], with the exception being that while there was a significantly higher proportion of control participants who were married or had a partner compared with the biomarker feedback intervention group, this difference did not remain statistically significant in MAW Phase 4 due to the smaller sample size.

### 3.2. Tobacco Use Patterns

Table 2 presents the tobacco use status at different time points beginning with the time of at delivery. Thirty women (94%) reported they had used tobacco during the first 12 months after delivery, of which 17 (57%) reported they had tried to quit at least once during this time period.

Of the 32 participants, 29 (91%) reported current tobacco use. Of these, 28 (97%) smoked cigarettes. Current smokers reported a mean of 6.8 CPD (SD = 5.4, range 0.5–20.0). Of the 29 current tobacco users, 22 (76%) used tobacco both at the time of delivery and during the first 12 months postpartum, and 6 (21%) did not use tobacco at delivery but used tobacco during the first 12 months postpartum.

### 3.3. Tobacco Relapse Rate

Nine women reported they did not use tobacco at delivery. Of these, seven (78%) reported using tobacco during the first 12 months after delivery and cigarette smoking was the predominant tobacco product used.

### 3.4. Social–Environmental Context

At the time of the interview, 20 (62%) participants reported that their spouse/partner currently used tobacco. Thirty-one (97%) participants reported a home smoking ban. The average number of hours exposed to cigarette smoke each day was 2.9 (SD = 4.2, range 0–15) hours.

### 3.5. Reasons for Postpartum Tobacco Use

Of the 32 participants, 27 (84%) reported stress and 15 (52%) indicated addiction as the reason why they had continued, started, or resumed tobacco use. Of participants mentioning “stress,” most did not elaborate. Illustrative quotes (noted in quoted text) or paraphrases from the interviews relevant to the theme of stress-related tobacco use included: (1) management of daily postpartum related stressors, e.g., “worrying about taking care of my baby and not getting enough sleep,” “it’s a stress relief,” “a break outside, time to myself,” and daughter born prematurely; (2) behavioral health issues, e.g., “anxiety, helps me from panicking, reduce stress,” grief from loss of a family member, depression, anxiety, and stress-coping mechanism; and (3) intimate partner violence and associated child welfare concerns, e.g., child taken away by social services, because she does not have her baby and has not seen her daughters in a long time, domestic violence, and abusive relationships.

For the theme of addiction, examples included: “I have tried so many times to quit smoking cigarettes but it’s really hard” and “mostly the nicotine I think, addiction” along with quotes related to duration of smoking, making it difficult to quit, e.g., “really hard to quit physically and mentally. Been using tobacco since I was nine years old. Accepted in the village.”

All 32 participants were asked about reasons why they thought a woman would start using tobacco again after delivery. The primary reason reported for 23 participants (72%) was to control stress. Other reasons were addiction reported by seven participants (22%), managing depression reported by five participants (16%), as well as not being pregnant anymore or breastfeeding indicated by five participants (16%).

### 3.6. Perceived Stress

The mean (SD) PSS score for the sample was 5.1 (3.6), range 0–14. Among the 29 current tobacco users, the mean score was 5.3 (3.7), range 0–14. The mean score overall (*p* = 0.35) or when including only current tobacco users (*p* = 0.25) was not significantly different than the mean reported for a U.S. representative sample (4.5).

### 3.7. Negative Affect

The average NA score for the sample was 18.7 (9.2), range 10–47. Among the 29 current tobacco users, the mean score was 19.4 (9.3), range 10–47. The mean score for both the overall sample (*p* = 0.023) and when including only current tobacco users (*p* = 0.013) was significantly greater than the mean reported for a U.S. representative sample (14.8).

## 4. Discussion

In this preliminary study, we assessed tobacco use postpartum along with reasons for continued tobacco use and smoking relapse among AN women who had previously enrolled in a smoking cessation intervention study during pregnancy. A key finding was the high proportion of continued tobacco use reported within the 12 months after delivery (94%) as well as one to three years postpartum (91%). Relapse to tobacco use among women who had quit using at the time of delivery was 78% compared to a general population sample of AN women from the Alaska Pregnancy Risk Assessment and Monitoring System (PRAMS) study (57%) [1]. However, the PRAMS study surveyed women an average of four months postpartum (range 2–9), a much shorter duration than in our study. From the 2017 Alaska Childhood Understanding Behavior Survey, a three year postpartum follow-up survey to PRAMS, the prevalence of any cigarette smoking in the last two years was 50% among AN women in comparison to 11% of Alaskan white women [27]. We also found that stress was a major reported reason for continuing, starting, or resuming tobacco use after delivery. Our study is the first to explore the reported reasons for tobacco use among postpartum AN women.

Consistent with our results, a systematic review of qualitative and quantitative studies conducted in the U.S. and other countries identified smoking for stress management as a key barrier to quitting smoking among Indigenous people and other vulnerable groups [28]. Internationally, in general population samples, it is well documented that smokers report greater levels of stress compared to non-smokers [29] and post-traumatic stress disorder has been associated with current cigarette smoking [30]. Moreover, analysis of the U.S. PRAMS study (2000–2011) found that factors associated with postpartum smoking were experiencing three to five stressors in pregnancy, not breast feeding, not having an in-home smoking ban, and having an unplanned pregnancy [11]. A systematic review of seven qualitative studies involving 250 Indigenous pregnant women in New Zealand and Australia found that stress and times of chaos within the women’s lives impacted on their capacity to prioritize smoking cessation [31].

Although participants in our study reported stress as a reason for tobacco use, most did not elaborate on the types of stressors they experienced. Along with stress associated with parenting, some examples provided indicated traumatic types of life events such as domestic violence, abusive relationships, and death of a family member. Analysis of the Alaska PRAMS (2004–2011) found postpartum AN women retrospectively reported more stressful life events during the 12 months prior to the birth of their baby compared to women of other races (*p* < 0.001) [32]. Stressors more commonly endorsed among AN women than non-Native women were death of someone close, arguing with husband/partner more than usual, having someone close with a substance use problem, and moving to a new address.

About half (52%) of the women in our sample reported addiction as a reason for their use of tobacco during the postpartum period. Cigarette smokers at the time of the interview reported a mean of 6.8 CPD. This is slightly higher than a study of pregnant AN women in our prior MAW Phase 1 study (mean 4.1 CPD) but lower than amounts reported in general population samples of pregnant women (i.e., mean 12.3 CPD) [33]. One study [34] found that AN adult men and women cigarette smokers reported a mean of 7.8 CPD whereas U.S. smokers averaged 15 CPD. However, Benowitz et al. [35] observed that the average plasma cotinine levels in AN smokers were comparable to the average plasma cotinine levels among the overall U.S. population.

This study has several limitations to consider when interpreting the results. While the overall response rate was good (68%), as a preliminary study, the sample size is small. Some sample characteristics may limit the generalizability to other AN women. All women were cigarette smokers during pregnancy and had enrolled in a cessation treatment study conducted in an urban area of Alaska. However, to be enrolled, the women did not have to be ready to quit. Nonetheless, we do not have a general population comparison group of postpartum AN women. Our study was also limited to the 47 of 60 from our prior MAW Phase 3 study who provided consent to be re-contacted for future research. Most of the baseline data reported had been collected for MAW Phase 3, about two years prior, and thus some of the characteristics may have changed at the time of interview for MAW Phase 4. We did not collect data on social determinants of health (e.g., Women, Infant and Child (WIC) program eligibility, low income status) that may have influenced perceived stress and tobacco use. The small number of non-tobacco users at follow-up precluded comparisons with tobacco users on PSS and NA scores or other variables. We did not collect comprehensive information on tobacco use during follow-up, i.e., the time period from 12 months postpartum to the time of the interview. Due to logistical reasons, we did not biochemically verify self-reported non-tobacco use. In our MAW Phase 1 study, among the 64 participants self-reporting non-tobacco use, 63 (98%) had urine cotinine concentrations <50 ng/mL, indicating non-use [15]. Furthermore, due to the difficulty in reaching women, for some participants the timing of the interview occurred two to three years postpartum, which may have impacted recall of tobacco use status during the first year after delivery.

Despite these limitations, our preliminary results have several practice and research implications for exploring ways to support non-tobacco use among postpartum AN women. Clinically, we have previously reported on the need to enhance pregnant women’s use of the SCF Quit Tobacco Program standard of care counseling, as the MAW Phase 3 study resulted in a relatively high quit rate at delivery (20%) in both arms [17]. This program includes behavioral counseling and access to pharmacotherapy. The current results emphasize the importance of health educators, obstetricians, pediatricians, and other providers continuing to assess tobacco use, promoting referrals to the SCF Quit Tobacco Program, and utilizing other cessation resources among postpartum AN women.

Recommended clinical practice guidelines for behavioral counseling during tobacco cessation focus on strategies for heavier smokers, as well as general stressors (e.g., late to a meeting) and strategies to reduce stress (e.g., taking a walk) [36]. Participants in our sample experienced stress on a spectrum from daily stressors (e.g., parenting) to unique traumatic events (e.g., death of a loved one) and long-term traumatic experiences (e.g., intimate partner violence). These results highlight that different messaging and strategies may need to be considered by providers for AN women to address the varying types of stressors they experience, such as meeting women where they are in terms of recognizing traumatic experiences [37]. AN people value family, community, and culture, all of which should be incorporated in cessation messages and strategies geared toward de-escalating these different types of stressors.

Recent reports documented the continued disparities in tobacco use prevalence and limited culturally-specific tobacco treatments developed for AI/AN women [38] as well as Indigenous women in other countries [14]. Research has also highlighted racial/ethnic disparities in perceived stress, tobacco use, and cessation in population-based and treatment samples [38,39], and among pregnant women specifically [37], but little is known about these associations in AI/AN women. Collectively, our formative work indicates that perceived stress is an important concern reported by both pregnant [12,13] and postpartum AN women that may help to explain known disparities in tobacco use and quitting in this population; this topic could be examined in future research.

Gould and colleagues [14] highlighted the need for qualitative studies with Indigenous women to understand cultural strengths supporting quitting tobacco in order to develop more effective approaches. Using a community-based participatory research (CBPR) [40] approach, future qualitative work could assess the types of stressors, as well as preferences for culturally relevant messaging and ways to de-escalate stress among AN women. This work could be done to understand the meaning of “stress” and the types of stressors women are dealing with on a daily basis (e.g., parenting). Qualitative work could also assess the meaning of “addiction” among AN women, when they typically smoke (time of day), and what stress or situations they are taking a break from to smoke. Future research could focus on AN women of childbearing age, with the goal of preventing tobacco use during pregnancy and the postpartum period.

The Indigenist Stress-Coping Model [41] could be a useful conceptual basis for future qualitative studies. This framework delineates how multiple cumulative and co-occurring stressors (i.e., experiences of historical trauma, loss of traditions, adverse childhood experiences, and traumatic life events in adulthood) among AI/AN women are linked to both communal and individual contemporary health and health behaviors, including tobacco use [42,43]. Analysis of data from the Adverse Childhood Experiences (ACEs) study found that, among Alaskans, the prevalence of four or more adverse ACEs was significantly higher among AN adults compared with non-native people (28% vs. 15%) [42]. ACEs have been linked to increased prevalence of tobacco use in adulthood [42,44]. The Indigenist Stress-Coping Model [41] also incorporates cultural and community resilience, including traditionalism and spiritual coping practices as moderators that may buffer the effects of stress [42,45].

## 5. Conclusions

In this sample of AN women, postpartum tobacco use was highly prevalent and stress was a primary reason that women endorsed for using tobacco. These preliminary results have several practice and research implications for exploring ways to support non-tobacco use among postpartum AN women. Foremost, external factors that evoke continuous stress among AN women and inhibit successful smoking cessation or continued abstinence should be identified, explored, and addressed. These will vary from person to person and most likely require utilization of outside resources to reduce external stressors such as financial barriers and/or prevailing behavioral and mental health burdens. Self-efficacy and personal beliefs surrounding postpartum tobacco use are important internal factors that should also be identified and, if needed, influenced to best support successful cessation or abstinence. Future research should include investigation of resiliency factors that helped pregnant women successfully quit using tobacco during pregnancy and continued abstinence postpartum.

## Figures and Tables

**Figure 1 ijerph-16-03024-f001:**
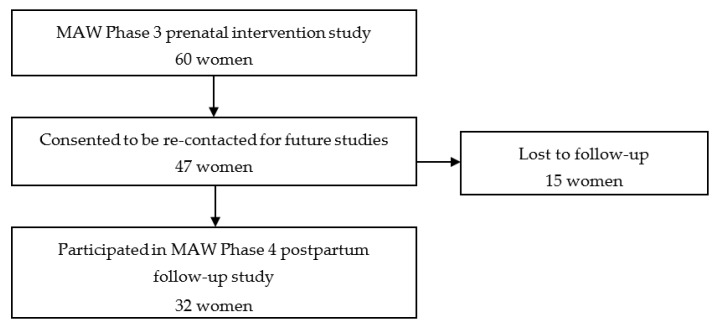
Participant flow.

**Table 1 ijerph-16-03024-t001:** Participant socio-demographic and tobacco use characteristics: MAW Phase 4 study ^1,2^.

Characteristic	Total (*n* = 32)
Study group	
Standard care control	18 (56)
Biomarker feedback intervention	14 (44)
Age	28.5 ± 3.9
Range	21–36
Married/partner	11 (34)
Education	
Less than high school	8 (25)
High school/GED	9 (28)
Some college	15 (47)
Number weeks gestation	13.6 ± 6.5
Range	5–28
One or more biological children	29 (93)
Spouse/partner smokes ^3^	9 (82)
Home smoking ban (includes artic entry)	31 (97)
Hours exposed to cigarette smoke each day	3.8 ± 3.2
Range	0–12
Lives with other smokers	24 (75)
Contemplation Ladder score	7.1 ± 2.0
Medium (4–6)	12 (37)
High (7–10)	20 (62)
Cigarettes smoked per day	4.6 ± 3.1
Range	1–13
FTCD total score ^4^	2.4 ± 2.0
Range	0–6
Urinary cotinine–creatinine corrected (ng/mg-creat)	
Median	561
Range	33.3–2839.0

^1^ Participant baseline characteristics obtained from the MAW Phase 3 study. ^2^ Reported as *n* (%) or mean ± standard deviation (SD) and range as appropriate. ^3^ For participants reporting a spouse/partner. ^4^ FTCD = Fagerström Test for Cigarette Dependence. Possible scores range from 0–10.

**Table 2 ijerph-16-03024-t002:** Percentage self-reporting tobacco use among postpartum Alaska Native women (*n* = 32).

Time Point *n* (%)			
At Delivery	During the First 12 Months after Delivery	At 12 Months Postpartum	Current Status (2 Years Postpartum)
23 (72%)	30 (94%)	27 (84%)	29 (91%)

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
