# Peer review of "Postpartum Tobacco Use and Perceived Stress among Alaska Native Women: MAW Phase 4 Study"

_ijerph, 2019, doi:10.3390/ijerph16173024_

Round 1
Reviewer 1 Report
Methods:
Whether you analyzed the use of e-cigarettes or a mixed ?
In my opinion it would be better to show the study sample in the flow chart (including Phase 1 to 4)
Results:
Standardize the method of writing data, eg: 62.5%(20/32) (line 205) vs. 27 (84%) (line 209) vs. 16%, n=5) (line 226).
In my opinion, in a small group, the percentage data can be misleading and is not very legible eg: 17 (56.7%) ... 28 (97%) (lines 194-196)
Author Response
Response to Reviewer 1
Methods:
Whether you analyzed the use of e-cigarettes or a mixed?
As suggested, we added under the Measures section the types of tobacco/nicotine products that were assessed, including use of e-cigarettes.
In my opinion it would be better to show the study sample in the flow chart (including Phase 1 to 4). We clarify in the Introduction section that MAW Phases 1-3 were conducted in different samples. Phase 4 participants were linked to MAW Phase 3 only. As recommended, we added a Figure to better show the study sample for which the Phase 4 interviews were conducted.
Results:
Standardize the method of writing data, eg: 62.5%(20/32) (line 205) vs. 27 (84%) (line 209) vs. 16%, n=5) (line 226). In my opinion, in a small group, the percentage data can be misleading and is not very legible eg: 17 (56.7%) ... 28 (97%) (lines 194-196).
Based upon comments on this issue from both reviewers, we revised the text to make the reporting of sample size numbers and percentages consistent, including removing decimals from some of the percentages reported. We retained the percentages for ease of interpretation, but added the number of participants for each percentage reported. Results were revised to be consistent throughout the manuscript, with the numbers reported first, then percentages.
Reviewer 2 Report
This study follows-up on a small sample of Alaska Native women 1-3 years postpartum and collected information on self-report tobacco use status. Overall, the paper is well written, and provides novel data on tobacco use during the postpartum period in a population that is commonly underrepresented in research and with few available culturally tailored interventions. The major limitation of the paper is the small sample size and the generalizability of the data to other American Indian/Alaska Native tribes. However, the authors are cognizant of this limitation (they thoroughly state it as a limitation in the discussion section and describe the results as preliminary). A major strength of this paper is that it can be cited by future research to document the important need for more culturally-tailored interventions among pregnant/postpartum AN women as well as AN in general. The following are minor suggestions that the authors should consider before publishing:
1. The journal’s audience is international. Therefore the background should discuss how this research may have implications for other Indigenous populations as well such as New Zealand Mari.
2. A comment on reasons for higher prevalence of tobacco use among pregnant AN would be helpful to the reader. Tobacco is used for traditional/ceremonial purposes in some Native Tribes. Was information collected on traditional tobacco in this population? At the least, the authors should describe in the Measures section on tobacco use under
3. What year were these studies conducted (MAW Phases). This should also be added to the abstract (the year of the Phase 4 study).
4. Why did the authors selected 1-3 years postpartum. This is a wide time window that could impact recall. Participants who recalled their tobacco use at 1 year postpartum may have had difficulty, especially if the interview was taken placed 3 years postpartum. This should be discussed as a limitation.
5. Given the small nature of the data, the authors should include sample size numbers in the abstract after the percentages are displayed. For example, “Tobacco use status was 72% (n=XX) at delivery…”
6. The statistical methods should be redone using non-parametric tests given the small sample sizes.
Author Response
Response to Reviewer 2
This study follows-up on a small sample of Alaska Native women 1-3 years postpartum and collected information on self-report tobacco use status. Overall, the paper is well written, and provides novel data on tobacco use during the postpartum period in a population that is commonly underrepresented in research and with few available culturally tailored interventions. The major limitation of the paper is the small sample size and the generalizability of the data to other American Indian/Alaska Native tribes. However, the authors are cognizant of this limitation (they thoroughly state it as a limitation in the discussion section and describe the results as preliminary). A major strength of this paper is that it can be cited by future research to document the important need for more culturally-tailored interventions among pregnant/postpartum AN women as well as AN in general.
Thank you for the positive feedback.
The following are minor suggestions that the authors should consider before publishing:
- The journal’s audience is international. Therefore the background should discuss how this research may have implications for other Indigenous populations as well such as New Zealand Mari.
Thank you for this recommendation. We have revised the Introduction and Discussion sections to add references to international work conducted more generally, and in other populations of Indigenous women.
- A comment on reasons for higher prevalence of tobacco use among pregnant AN would be helpful to the reader. Tobacco is used for traditional/ceremonial purposes in some Native Tribes. Was information collected on traditional tobacco in this population? At the least, the authors should describe in the Measures section on tobacco use under.
We added to the Introduction section that prior studies documented that tobacco is not used for traditional (ceremonial or religious) purposes among Alaska Native people.
- What year were these studies conducted (MAW Phases). This should also be added to the abstract (the year of the Phase 4 study).
As suggested, we added this information to the first paragraph of the Methods section.
- Why did the authors selected 1-3 years postpartum. This is a wide time window that could impact recall. Participants who recalled their tobacco use at 1 year postpartum may have had difficulty, especially if the interview was taken placed 3 years postpartum. This should be discussed as a limitation.
Thank you for pointing out this limitation. We clarify in the Methods section that study staff attempted to contact women within one to two years after delivery. However, after several attempts, study staff was unable to contact several women, and made multiple attempts to reach women who eventually participated. This resulted in a large time window (1-3 years postpartum) than planned for assessing current tobacco use status. We note this as a limitation, potentially impacting recall of tobacco use status, in the Discussion section.
- Given the small nature of the data, the authors should include sample size numbers in the abstract after the percentages are displayed. For example, “Tobacco use status was 72% (n=XX) at delivery…”
As recommended, changes were made to the abstract to include the sample size numbers. Also, see response to reviewer 1 above.
- The statistical methods should be redone using non-parametric tests given the small sample sizes.
As recommended, we used a non-parametric (two-sample rank sum) test for continuous variables when comparing the baseline characteristics between study groups. The p-values remained not significant when using a two-sample rank sum test.
For the Perceived Stress Scale (PSS) and Negative Affect (NA) scale scores, we do not have access to the normative raw data for conducting a one sample non-parametric test. Thus, we used the one-sample t-test for comparing our data on PSS and NA scores to the mean scores reported from the normative data.